# The Angiography Pattern of Buerger’s Disease: Challenges and Recommendations

**DOI:** 10.3390/jcm14144841

**Published:** 2025-07-08

**Authors:** Bahare Fazeli, Pavel Poredos, Aaron Liew, Edwin Stephen, Abul Hasan Muhammad Bashar, Matija Kozak, Mariella Catalano, Mussaad Mohammaed Al Salman, Louay Altarazi, Pier Luigi Antignani, Sanjay Desai, Evangelos Dimakakos, Dilek Erer, Katalin Farkas, Marcus Fokou, Emad Hussein, Mihai Ionac, Takehisa Iwai, Oguz Karahan, Daniel Kashani, Albert Kota, Knut Kroger, Emre Kubat, Prabhu Prem Kumar, Rafal Malecki, Antonella Marcoccia, Sandeep Raj Pandey, Malay Patel, Zsolt Pecsvarady, Adil Polat, Hassan Ravari, Gerit Schernthaner, Dheepak Selvaraj, Hiva Sharebiani, Agata Stanek, Andrzej Szuba, Wassila Taha, Hossein Taheri, Hendro Sudjono Yuwono, Mustafa Hakan Zor

**Affiliations:** 1Support Association of Patients of Buerger’s Disease (Buerger’s Disease NGO), Mashhad 9183785197, Iran; bahar.fazeli@gmail.com; 2Department for Vascular Diseases, Medical Faculty of Ljubljana, University Medical Center Ljubljana, 1000 Ljubljana, Slovenia; matija.kozak@kclj.si; 3Department of Medicine, Portiuncula University Hospital, Saolta University Healthcare Group, H53 T971 Galway, Ireland; aaron.liew@nuigalway.ie; 4School of Medicine, University of Galway, H91 TK33 Galway, Ireland; 5Vascular Surgery Department, Sultan Qaboos University Hospital, Muscat 123, Oman; edwinmay2013@gmail.com; 6National Institute of Cardiovascular Diseases & Hospital (NICVD), Dhaka 1207, Bangladesh; ahmbashar@gmail.com; 7Department of Biomedical and Clinical Sciences, Inter-University Research Center on Vascular Disease, L. Sacco Hospital, University of Milan, 20157 Milan, Italy; mariella.catalano@unimi.it; 8Division of Vascular Surgery, King Saud University, Riyadh 11451, Saudi Arabia; mussaad@ksu.edu.sa; 9Varicose Veins and Vascular Polyclinic (VVVC), Damascus 11956, Syria; dr.l.tarazi@gmail.com; 10Vascular Center, Nuova Villa Claudia, 00100 Rome, Italy; plantignani@gmail.com; 11Department of Vascular and Endovascular Surgery, Ramaiah Medical College Hospital, Bangalore 560054, India; scdesai@hotmail.com; 12Department of Internal Medicine, University of Athens Public Hospital, SOTIRIA, 11527 Athens, Greece; edimakakos@yahoo.gr; 13Independent Researcher, Ankara 06510, Turkey; dilekerer@hotmail.com; 14Department of Angiology, St. Imre University Teaching Hospital, 1115 Budapest, Hungary; farkask@hotmail.com; 15Department of Vascular and Kidney Transplantation Surgery, Yaounde General Hospital, Yaounde P.O. Box 5408, Cameroon; mfokou@yahoo.com; 16Vascular Surgery Department, Ain Shams University, Cairo 11351, Egypt; emad.hussein52@hotmail.com; 17Vascular Surgery and Reconstructive Microsurgery, Victor Babes University of Medicine and Pharmacy, Piata Eftimie Murgu 2, 300041 Timisoara, Romania; mihai.ionac@gmail.com; 18Tsukuba Vascular Center, Moriya City 302-0118, Ibaraki Prefecture, Japan; iwai@keiyu.or.jp; 19Department of Cardiovascular Surgery, Medical School of Alaaddin Keykubat University, Alanya 07400, Turkey; oguzk2002@gmail.com; 20Division of Hospital Medicine, Mayo Clinic Florida, Jacksonville, FL 32224, USA; kashani.daniel@mayo.edu; 21Vascular Surgery Unit, Flinders Medical Centre, Adelaide 5042, Australia; albert.cmc@gmail.com; 22Department of Vascular Medicine, HELIOS Klinik Krefeld, 47805 Krefeld, Germany; knut.kroeger@helios-gesundheit.de; 23Department of Cardiovascular Surgery, Gülhane Training and Research Hospital, Health and Science University, Ankara 06010, Turkey; ekubat@gmail.com; 24Department of Vascular Surgery, Christian Medical College, Vellore 632004, India; prabhupremkumar@yahoo.co.in (P.P.K.); drdheepak@cmcvellore.ac.in (D.S.); 25Faculty of Medicine, Wrocław University of Science and Technology, Hoene-Wroński 13c, 58-376 Wrocław, Poland; rmalecki@gazeta.pl; 26Angiology and Autoimmunity Medical Unit, Rare Diseases Reference Center for Systemic Sclerosis, Sandro Pertini Hospital, 00157 Rome, Italy; antonella.marcoccia@aslroma2.it; 27Vascular and Endovascular Surgery Department, Annapurna Hospital, Kathmandu 44600, Nepal; sandeeprajpandey@gmail.com; 28Vascular Surgery Department, Apollo-CVHF Hospital, Ahmedabad 380009, India; info@drmalaypatel.com; 29Department of Vascular Medicine, Flor Ferenc Teaching Hospital, 2143 Kistarcsa, Hungary; pecsvarady@gmail.com; 30İstanbul Bagcilar Research and Training Hospital Cardiovascular Surgery, University of Health Sciences, Istanbul 34200, Turkey; dradilpolat@gmail.com; 31Vascular Surgery Research Center, Emam Reza Hospital, Mashhad University of Medical Sciences, Mashhad 9137913316, Iran; ravarih@mums.ac.ir; 32Division of Angiology, Department of Internal Medicine II, Medical University of Vienna, 1090 Vienna, Austria; gerit.schernthaner@me.com; 33Department of Philosophy, School of Humanities, University of Auckland, Auckland 1010, New Zealand; hivasharebiani@yahoo.com; 34Department of Internal Medicine, Angiology and Physical Medicine, Faculty of Medical Sciences in Zabrze, Medical University of Silesia, 40-635 Bytom, Poland; agata.stanek@gmail.com; 35Department of Angiology and Internal Medicine, Wroclaw Medical University, Borowska 213 str, 50-556 Wrocław, Poland; szubaa@yahoo.com; 36Non Invasive Vascular Lab, Al Salam Hospital, Cairo 12411, Egypt; wassila.taha@gmail.com; 37Department of General Surgery, Farabi Hospital, Mashhad 9178686917, Iran; ht23766@yahoo.com; 38University Islam Bandung, Bandung 40116, Indonesia; hsyabc47@gmail.com; 39Department of Cardiovascular Surgery, Faculty of Medicine, Gazi University, Ankara 06560, Turkey; mhzor76@gmail.com

**Keywords:** Buerger’s disease, thromboangiitis obliterans, angiography, arteriography, peripheral arterial disease, vasculitis, diagnosis

## Abstract

In 2023, the VAS international working group on Buerger’s Disease (BD) recommended two diagnostic criteria based on a prior Delphi study: “definitive” and “suspected”. The “definitive” criteria are history of smoking, typical angiography, and typical histopathological features. All three features are mandatory to confirm a “definitive” diagnosis of BD. The conundrum is—what features should be considered typical of BD angiography? According to this review, segmental occlusion of infrapopliteal arteries, corkscrew collaterals that appear to continue the occluded arterial segment (Martorell’s sign) or bypass the segmental occlusion, absence of atherosclerotic plaque or aneurysm could differentiate BD from ASO. Hence, for “typical” BD angiography, these manifestations should certainly be considered. However, data for differentiating angiography patterns of BD from the small- and medium-sized vasculitis including Behcet’s disease, scleroderma, hepatitis associated vasculitis, and anti-phospholipid syndrome are limited. Further studies for investigating the angiography pattern in BD patients in early and late presentation of BD, particularly in the patients with long-term follow up, are highly recommended.

## 1. Introduction

Buerger’s disease (BD) is a relapsing–remitting segmental, inflammatory, thrombotic occlusive, and peripheral vascular disease with unknown etiology which usually involves the medium- and small-sized neurovascular bundle of predominantly young male smokers [1]. Recurring and sharp segmental inflammation and intraluminal thrombus, along with endothelial cell proliferation, may lead to the occlusion of small- to medium-sized arteries in the extremities, causing tissue gangrene or limb loss.

Exacerbation and prognosis of BD have been shown to be closely related to tobacco smoking [1]. However, the etiological role of tobacco smoking in BD development is challenging; because it is unknown why, amongst millions of smokers, only a small number develop BD. It also remains unknown why, of the BD patients who continue smoking, only half of them show aggravation and undergo amputations. Therefore, the exact etiology of BD is unclear [1].

BD has a geographical distribution and it is more prevalent in the Middle East, Far East, Southeast Asia, and Southern America in comparison with Northen America and Western Europe [1]. Notably, the geographical distribution of BD has changed in the past decades and in some countries like Japan, Thailand, and USA, the prevalence of BD has declined without any reasonable changes in the prevalence of smokers [1].

Until recently, there was no definitive biomarker for BD diagnosis due to its unknown etiology. Several diagnostic criteria have been suggested for BD diagnosis [2]. In the countries that BD is more common, the diagnosis is usually made based on clinical features of the disease with or without investigating the common laboratory risk factors for ASO. However, in the regions that BD seems to be a rare disease, the diagnosis is made by ruling out other types of vascular diseases and vasculitis using imaging and specialized laboratory tests.

BD is usually considered a rare disease. Therefore, multi-central studies are very important for better understanding the nature of BD and consequently finding a better approach for BD treatment.

However, before any global studies are conducted, data harmonization among the centers for BD diagnosis is essential. On the other hand, diagnostic criteria that could be acceptable in any region despite the prevalence of BD should be the first step.

In 2023, the international working group on BD of VAS independent foundation in Angiology/Vascular Medicine by 28 countries from different regions recommended two diagnostic criteria based on a prior Delphi study: “definitive” and “suspected”. The “suspected” criteria could be used in daily practice for screening, early diagnosis, and data harmonization on the prevalence or treatment of BD. The “suspected” criteria includes history of tobacco smoking with four out of five findings of disease onset being less than 45 years, ischemic involvement of both lower limbs, ischemic involvement of any of the upper limbs, thrombophlebitis migrans, and Buerger’s color on edematous toes/fingers.

However, the “definitive” diagnosis can be made in a patient with atypical clinical presentation (the age of disease onset being more than 50 years, suffering from diabetes or hyperlipidaemia) or when we want to link a laboratory finding or an unreported clinical presentation to Buerger’s disease. The “definitive” criteria are history of smoking, typical angiography, and typical histopathological features. All three features are mandatory to confirm a diagnosis of BD [3].

Therefore, the next steps of our investigation are what features are the most important and should be considered as typical BD angiography and typical histopathology?

In this review, we evaluate the angiography patterns of lower/upper limbs, and visceral vessels in patients with a BD diagnosis and atypical cases of BD from 1953 to recent times to discuss the main features necessary to confirm a diagnosis of BD based on the frequency of the reported findings.

### 1.1. Angiography Pattern of Lower Limbs in BD Patients

Leo Buerger attempted to differentiate BD from atherosclerosis obliterans (ASO) in 1953 [1]. Although BD is an inflammatory vascular disease, there are fewer challenges in differentiating BD from other types of vasculitis. Several publications address the conundrum of whether BD is a distinct entity or a premature ASO [4,5,6,7]. Several findings have been addressed in BD angiography since 1953 to date:A.Normal aortoiliac and femoropopliteal arteries

Several studies have emphasized the absence of atherosclerotic plaque and aneurysms in proximal arteries in the angiography of BD patients, particularly in the non-affected limb [1,8,9,10]. This absence of atherosclerotic plaque in the involved limb has been found in 90% of the patients with a diagnosis of BD; besides this, abrupt occlusion of the femoropopliteal segment has been found in 40% of the patients in addition to infrapopliteal arterial occlusion [11,12].

The occlusion of iliac arteries in addition to infrapopliteal arterial occlusion (Figure 1) has also been reported in BD patients [13,14,15,16]. BD diagnosis for those patients with atherosclerotic lesions, iliac or femoropopliteal occlusion in arteriography was based on clinical manifestation and histopathology features of arteries harvested from amputees [1,11,12,17].

The literature has questioned the involvement of proximal arteries as being secondary to early stages of atherosclerosis due to the progression of BD OR proximal arteries by intima thickening and inflammatory processes in the wall of the large vessel before complete occlusion [4,5]. However, normal vessels without any irregularities in non-affected limbs of BD patients have been noticed [1,11,12,17].

In short, it has been emphasized that the presence of obliterative lesions in the infrapopliteal arteries, with normal femoral or iliac arteries, is a specific finding of BD. However, the involvement of proximal arteries may not exclude a BD diagnosis.

B.Segmental occlusions

In BD patients, the multi-segment occlusion of the infrapopliteal arteries and veins, so-called “skip lesions”, has been noticed in angiography [13,14,15,16,17,18,19]. Notably, the peroneal artery seems frequently patent, and it is usually involved less severely and less frequently as compared to the anterior tibial and posterior tibial arteries [20,21]. (Figure 2)

Also, segmental vasoconstriction had also been reported, which led to occlusion within two years [22]. Segmental occlusions in the femoropopliteal level appear to be rare [18,23]. Since arterial lesions in ASO are diffuse rather than segmental, this finding has been highlighted to differentiate BD from ASO [4,17,18,23].

C.Collaterals

In 1951, Martorell described direct tortuous small vessels that followed the course of the thrombosed artery without any visualized termination and a no-refill phenomenon of the original vessel in the angiography (Figure 3). He implied that his finding was pathognomonic for BD [1,24,25]. Later, Mckuscik in 1962 emphasized abundant tortuous collaterals in these patients and described them as “corkscrew” and “spider legs” or “tree roots” [26]. In further studies, it was noticed that corkscrew-like collaterals could bypass the segmental occlusions (so-called corkscrew channels) or directly follow the occluded artery (Martorell’s sign). Also several corkscrew-like collaterals beyond the popliteal occlusion (tree roots) usually observed in distal arteries [1,6,12,15,16,17,19,27,28,29,30] (Figure 4).

Several concepts attempting to explain corkscrew formation have been proposed by angiologists, moving away from the categorization of corkscrew collaterals. Some consider tortuous collaterals with large helical signs as corkscrews while some consider only the tortuous collaterals with tiny helical signs [28]. Maybe, that is the reason that some authors consider corkscrews as one of the criteria for BD diagnosis, while others imply that observing corkscrew collaterals alongside the clinical manifestation of the patient and the absence of atherosclerotic plaque in angiography could support BD diagnosis because corkscrews may also be present in diabetes, atherosclerosis, repetitive embolization (Figure 5), scleroderma, and lupus erythematosus [4,31,32,33,34].

In addition to the challenges on the value of corkscrew collaterals in BD diagnosis, corkscrews have been important for a better understanding of BD pathophysiology (from 1962 to 1980). In the past, corkscrews were considered as thrombosis recanalization of the occluded vessel [17,26,27,28]. Suzuki in 1982 emphasized that corkscrews were dilated vasa vasorum of the occluded main artery and not recanalization of its occluded portion [11]. However, Bas reported corkscrew collaterals at the knee level of BD patients to originate from the vasa nervorum of the tibial nerve rather than the vasa vasorum of the occluded popliteal artery [35]. Nowadays, Martorell’s sign is losing its significance.

D.Vasospasm

Diffuse arterial narrowing and generalized vasospasm have been observed in about 15% of angiography in BD patients [11,12,36] (Figure 5). Segmental vasoconstriction of the arteries about two years before their occlusion has been reported [37]. Most authors believe that circular vasoconstriction has been induced by irritation from the contrast medium. However, Kohler implied that this appearance might be due to diffuse intimal thickenings and not the consequence of vasoconstriction [38].

Moreover, several ring-like circular segments of stenosis, like goose trachea in the process of arteriography during the 1960s and 1970s were reported and the sensitivity of the arteries of BD patients to the catheter was addressed as its reason (Figure 6). Since goose trachea appearance has been rarely reported in ASO, it was concluded that the arteries of BD patients are more sensitive to stimuli than ASO patients [3,4,8,14].

E.Early venous opacification

Early venous opacification seen on angiography in about 20% of BD patients is attributed to arteriovenous shunting in a few studies [11,12].

### 1.2. Phlebography of Lower Limbs in BD Patients:

In 1976, Chopra evaluated the phlebography of this subset and reported segmental occlusion of major deep veins, diffusely narrow small veins, collaterals, and irregular tortuosity of the veins [39].

### 1.3. Angiography Pattern of Upper Limbs of BD Patients:

The most commonly reported findings are occlusions of the ulnar or radial artery, or both, at or above the wrist, tortuous collaterals, but not corkscrews, in particular around the wrist and interphalangeal joints (Figure 7) without any evidence of atherosclerotic plaque [7,12,40,41,42,43].

### 1.4. BD Diagnosis According to Angiography in Atypical Cases:

Atypical cases of BD are usually individuals with risk factors of ASO (age > 50 years, diabetes, and dyslipidaemia) or in non-smokers. Absence of atherosclerotic plaque in the proximal arteries, abrupt occlusion, segmental occlusions of infrapopliteal arteries, and tree-root collaterals (but not corkscrews) are the most common findings seen on the angiography of such cases [44,45,46,47,48,49,50,51].

### 1.5. Angiography Pattern of Visceral Vessels in the Patients with Previous BD Diagnosis:

According to the case reports of visceral involvement in BD patients, the angiography pattern of visceral vessels is usually not as typical as that of the lower extremities’ corkscrew collaterals around the occluded vessel with less tortuosity compared to the lower extremities, and abrupt occlusion of the mesenteric or coronary arteries has been reported without any evidence of atherosclerosis. In most of the case reports, stenosis, multiple filling defects or total occlusion of the involved artery without corkscrew collaterals (Figure 8) have been noticed [6,15,18,22,52,53,54]. Therefore, differentiating BD from ASO according to such stenosis appears very difficult and in many cases such stenosis was considered as due to atherosclerotic plaque. In particular, atherosclerotic lesions of the visceral vessels have been identified via the autopsy of the patients with BD diagnosis [4,6,18]. Interestingly, Hong et al. in 2005 demonstrated that the stenosis of the coronary artery of a BD patient completely resolved in the follow-up angiography after two months without any intervention [51].

## 2. Discussion

There is no consensus on whether the angiography pattern of BD is diagnostic [55]. Some authors have implied that the angiographic patterns of BD are non-pathognomonic; they have concluded that the diagnosis should be established alongside the clinical course of the patient [31,32,33]. However, based on this concept, BD diagnosis in patients with atypical clinical presentations would be challenging.

The next challenge is the definition of “typical” angiographic pattern for BD patients. For years, the typical BD angiographic pattern has been described as normal-proximal arterial structure, absence of atherosclerotic plaque, lack of aneurysm, infrapopliteal arterial occlusion, corkscrew collaterals, and skip lesions [1,8,9,10]. However, the duration between the disease onset and performing the angiography could influence the pattern(s) seen. For instance, corkscrew collaterals may not be completely developed in the early manifestation of the disease [28,29,30,31,32,33]; in a patient suffering from BD for years, the involvement of femoropopliteal arteries or even atherosclerotic plaques in the affected limb could be observed [1,11,12,13,14].

Unfortunately, the literature about angiographic patterns in early presentation of BD or in patients with long-term suffering from the disease are limited. Therefore, we need a functional or mechanistic explanation of the angiographic pattern of BD alongside the statistical data from different studies to be able to define “typical” angiographic pattern.

Segmental occlusions of the infrapopliteal arteries, so-called “skip lesions”, have been reported in BD [1]. In 1984, Hagen reported multiple segmental involvement of infrapopliteal arteries in all his 44 patients [12]. This appears to be due to segmental intima thickening or segmental scar tissue surrounding the infrapopliteal arteries. Notably, segmental vasoconstriction of the arteries about two years before their occlusion has been noticed [22]. Multiple occlusions in diabetic patients suffering from ASO, but not the segmental occlusions, have been observed [13,14,15,16,17,19].

Corkscrew collaterals in BD are usually less than 1.5 mm and they follow the course of the original vessel without refilling the vessel (Martorell’s sign) or they bypass the segmental occlusions of infrapopliteal arteries. Corkscrew collaterals have also been reported in ASO. However, in ASO, the small corkscrew collaterals (less than 1.5 mm) have been observed around patent arteries [1,28].

Diffuse arterial narrowing and generalized vasospasm have been observed in about 15% of the angiography of BD patients, which might be due to inflammation of sympathetic ganglia [11,12,36]. However, we could not find any report of generalized vasospasm in ASO. Therefore, generalized vasospasm could be supportive of a BD diagnosis.

According to this review, atherosclerotic plaque has been reported in about 10% of the angiography of BD patients when their disease had been confirmed by histopathology of arterial lesions [11,12]. Therefore, absence of atherosclerotic plaque would be supportive for BD diagnosis but the existence of atherosclerotic plaque alongside clinical and other arteriography features of BD would not exclude it.

Although abrupt occlusion has been emphasized as one of the characteristics of BD angiography, it depends on the stage of the disease. Also, this feature could also be observed due to arterial emboli [11,12].

An aneurysm in the peripheral arteries of the lower limbs has been reported in only one case report according to our literature review [54]. The absence of an aneurysm could be supported by the intact internal elastic lamina of vascular wall in BD histopathology [1,8]. Therefore, the absence of an aneurysm is supportive for BD diagnosis.

## 3. Conclusions

Until recently, ASO has been the prominent differential diagnosis of BD. Such an attempt to differentiate BD from ASO might be due to different treatment approaches of these peripheral vascular diseases. For instance, ASO patients usually benefit more from angioplasty procedures and statins than BD patients. Also, ASO could be predictable in patients with risk factors of atherosclerotic plaque formation. The atherosclerotic risk factors could also be controlled more appropriately for preventing or managing ASO. However, a small number of smokers might develop BD and complete smoking cessation for heavy smokers is not an easy process.

Usually for BD diagnosis, the typical clinical presentation of BD alongside ruling out the presence of atherosclerotic plaque by either imaging such as duplex or CT angiography has appeared enough [18,56,57].

However, in BD cases with risk factors of atherosclerosis but not smoking, a typical angiography could be helpful for differentiating BD from ASO.

According to this review, segmental occlusion of infrapopliteal arteries, corkscrew collaterals that appear to continue the occluded arterial segment (Martorell’s sign) or bypass the segmental occlusion, absence of atherosclerotic plaque or an aneurysm could differentiate BD from ASO. Hence, for “typical” BD angiography, these manifestations should certainly be considered.

This “typical” BD angiography has been suggested based on the frequency of angiography findings in the literature from 1953 to recently, and it received the consensus of the VAS BD working group.

However, according to this review, there are some more points that should be considered.

First of all, atherosclerotic plaque in the affected limb of a BD patient, in particular in the patients with long-term clinical manifestation of BD might not rule out a BD diagnosis.

Moreover, most of the studies about BD angiographic pattern were conducted on the patients who had been reported to have critical limb ischemia. There are a lack of studies about BD angiographic patterns in early clinical presentation of BD. Perhaps, investigating the angiography pattern in early and late presentation of BD, particularly in the patients with long-term follow up could help us in better understanding BD pathophysiology.

Although BD is an inflammatory vascular disease, small- and medium-sized vasculitis are less considered as the differential diagnosis of BD. Unfortunately, data for differentiating the angiography pattern of BD from small- and medium-sized vasculitis including Behcet’s disease, scleroderma, hepatitis associated vasculitis, and anti-phospholipid syndrome are limited.

All these lacunae and queries leave room for ongoing research and need for collaborative efforts by treating physicians/surgeons/angiologists all over the world, especially from regions where the disease is more frequently seen.

## Figures and Tables

**Figure 1 jcm-14-04841-f001:**
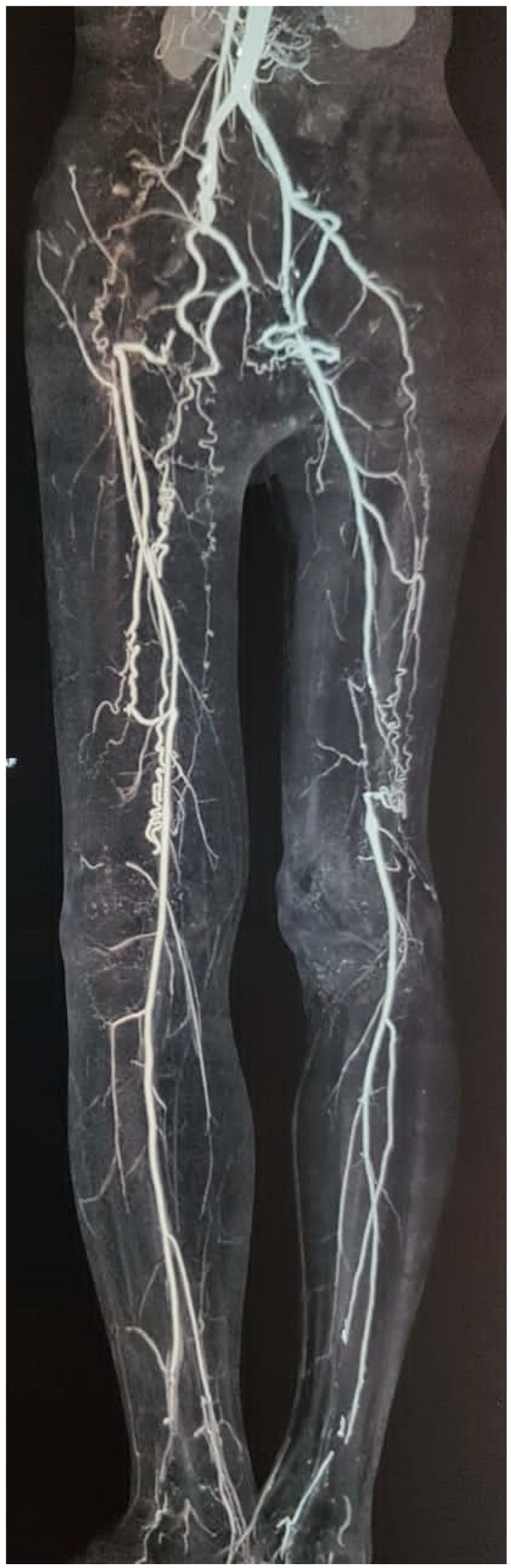
CT angiography of a 32-year-old male patient with history of smoking and clinical diagnosis of Buerger’s disease. Occlusion of iliofemoral arteries in addition to infrapopliteal arterial occlusions have been observed (courtesy of Abul HM Bashar).

**Figure 2 jcm-14-04841-f002:**
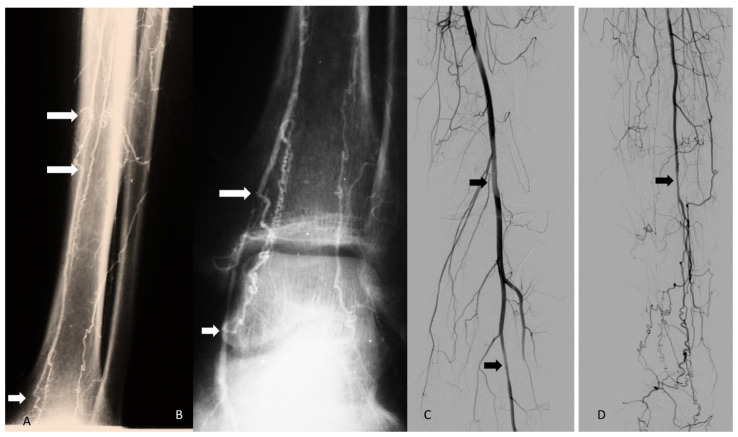
Segmental occlusions (**A**,**B**) and segmental stenosis (**C**,**D**) of infrapopliteal arteries have been bypassed by corkscrew collaterals (courtesy of B Fazeli and K Kroger). Arrows: All of them are showing segmental occlusions. But in the pictures with black background we had to use white arrows and in the gray background we had to use black arrows.

**Figure 3 jcm-14-04841-f003:**
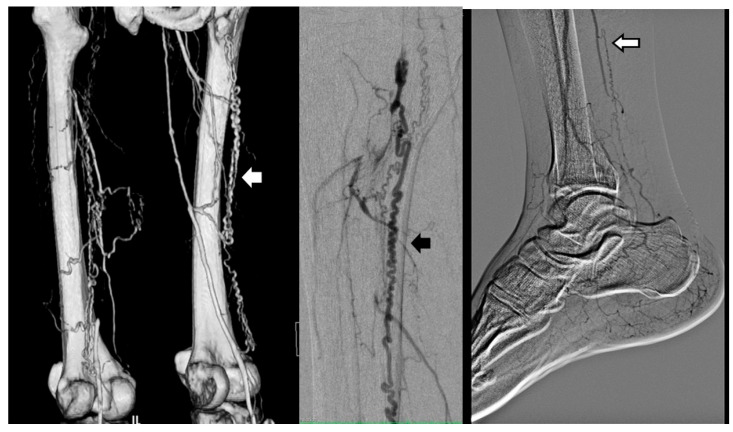
Martorell’s sign which might be enlarged vaso-nervorum (courtesy of B Fazeli and A Szuba). Arrows: All of them are showing segmental occlusions. The colors of the arrows are because of the color of the background.

**Figure 4 jcm-14-04841-f004:**
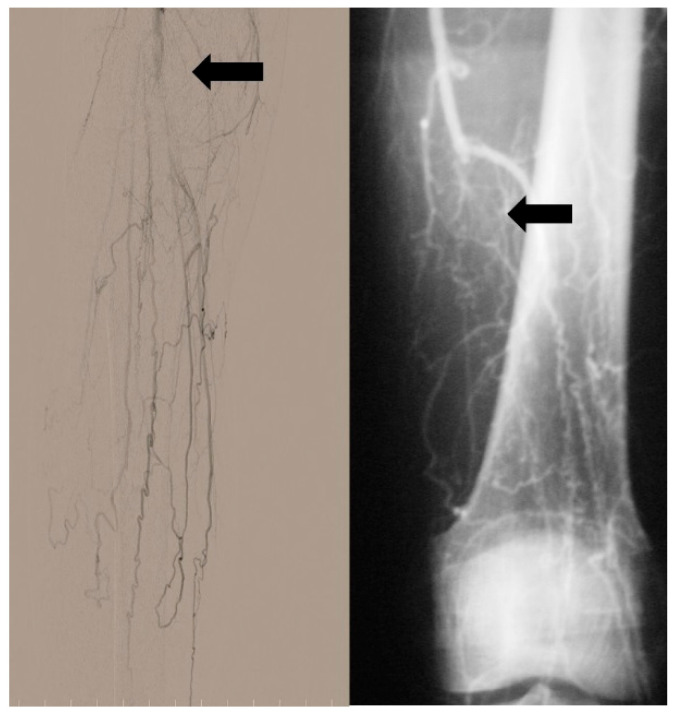
Tree root collaterals (courtesy of B Fazeli). The arrows show tree root collaterals.

**Figure 5 jcm-14-04841-f005:**
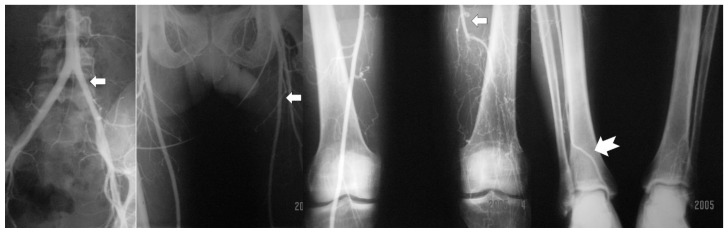
Vasospasm from its origin to occlusion of distal SFA. Notably, on the left side, the hypertrophied peroneal artery connects distally to the posterior tibial artery (courtesy of B Fazeli). The three pictures from the left side (arrows) shows vaso-spasm. The picture in the right side (arrow) shows the hypertrophied peroneal artery connects distally to the posterior tibial artery.

**Figure 6 jcm-14-04841-f006:**
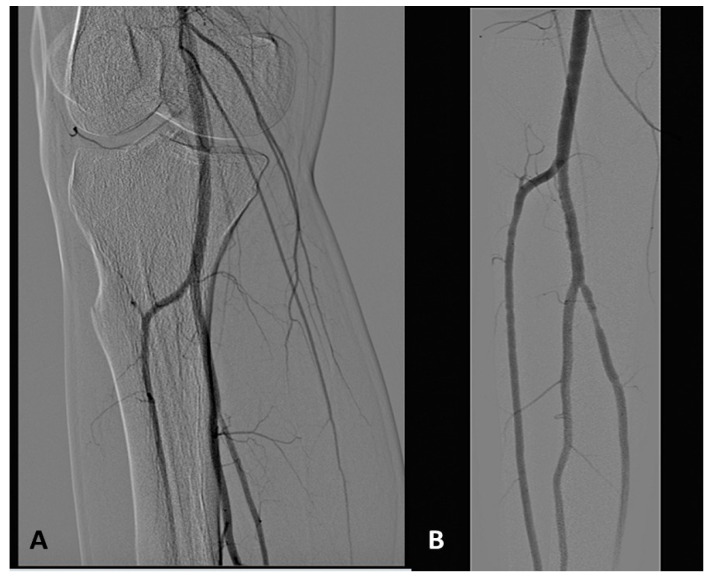
Angiography was performed for a 26-year-old man with clinical manifestation of BD and toe gangrene (**A**). Three days after the initial angiography, the patient underwent angioplasty of the distal part of posterior tibialis artery, and during the procedure, several ring-like circular segments of stenosis, like goose trachea, happened (**B**) (courtesy of B Fazeli).

**Figure 7 jcm-14-04841-f007:**
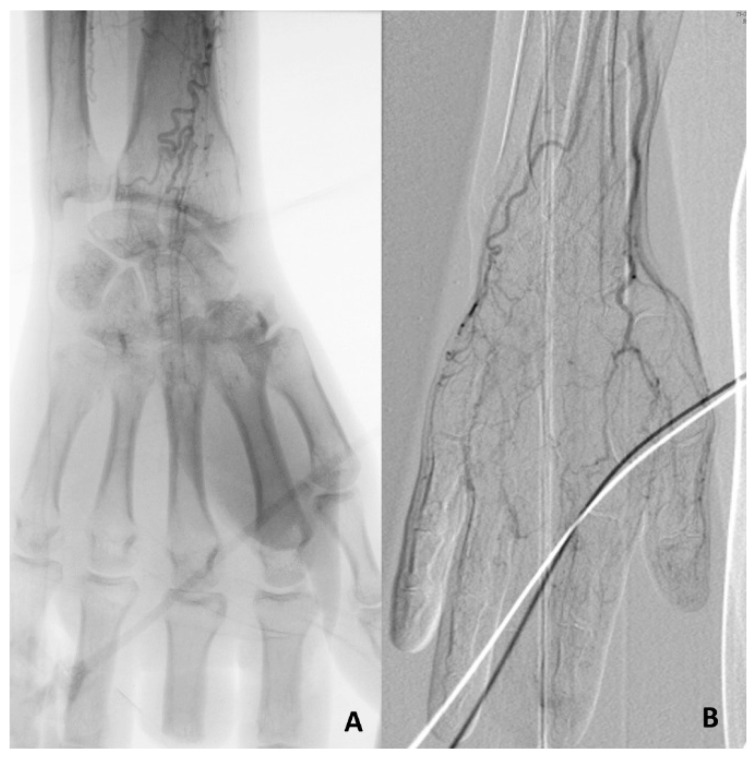
Angiography of upper limbs. Figure (**A**): Occlusion of both the radial and ulnar artery with sparse collateral circulation. Figure (**B**): Occlusion of the ulnar artery and arteries of the hand with tiny collateral vessels. The radial and the anterior interosseous are patent (courtesy of R Malecki).

**Figure 8 jcm-14-04841-f008:**
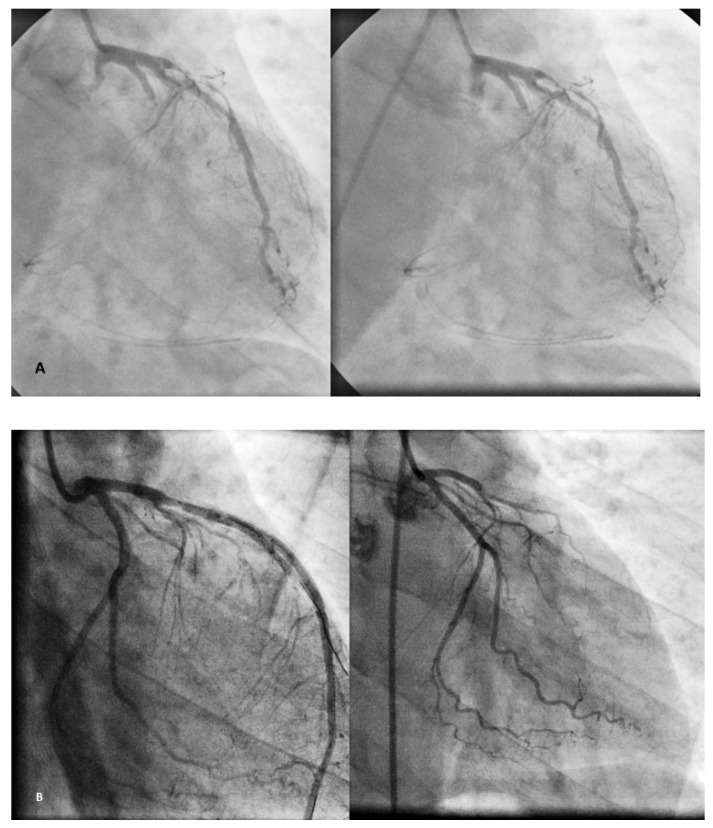
Atherosclerotic three-vessel coronary disease in a case of Buerger’s disease, twenty-one years after the disease onset (**A**). No significant stenosis in a 27-year-old BD male patient with chest pain two years prior of toe gangrene (**B**) (courtesy of B Fazeli).

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
