# Peer review of "The Angiography Pattern of Buerger’s Disease: Challenges and Recommendations"

_jcm, 2025, doi:10.3390/jcm14144841_

Round 1
Reviewer 1 Report
Comments and Suggestions for Authors
The authors have provided a systematic summary and review of the pathological characteristics of BD, which holds significant clinical importance and offers valuable reference material for clinicians in diagnosing and differentiating BD from ASO. I have the following questions for the authors' consideration:
-
In the Introduction, the authors offer an overview of the epidemiology of BD, specifically noting its varying incidence across different regions. However, this section relies solely on descriptive terms like "common," "uncommon," and "rare." Based solely on the text, it is difficult for the reader to discern the actual quantitative scale implied by these terms. Given that this is a review article, I believe this section should, at a minimum, present the approximate numerical ranges or magnitudes of these incidence rates.
-
Also within the epidemiological description in the Introduction, it is stated that the reasons for the significant differences in incidence rates across regions and ethnic groups "remain unexplained." Are there any current hypotheses or speculative explanations for these observed disparities?
-
The letter labels in the figure legends are too small, hindering readability. The quality of the figures and tables, in their current form, needs improvement.
-
A substantial portion of the article summarizes the differential diagnosis between BD and ASO. Could the authors elaborate on how distinguishing between BD and ASO specifically based on pathological characteristics informs or influences clinical management decisions and treatment strategies?
Author Response
- In the Introduction, the authors offer an overview of the epidemiology of BD, specifically noting its varying incidence across different regions. However, this section relies solely on descriptive terms like "common," "uncommon," and "rare." Based solely on the text, it is difficult for the reader to discern the actual quantitative scale implied by these terms. Given that this is a review article, I believe this section should, at a minimum, present the approximate numerical ranges or magnitudes of these incidence rates.
Thanks a lot for this great comment. I have searched about the quantification description of the epidemiological terms as “common”, “uncommon”, and “rare”. However, there is only one quantified description about “rare” term which means less than 1/2000 or 5/10000 cases. The rest of terms are only qualified and is going to be used for comparing. For this reason, the epidemiology part has been revised.
- Also within the epidemiological description in the Introduction, it is stated that the reasons for the significant differences in incidence rates across regions and ethnic groups "remain unexplained." Are there any current hypotheses or speculative explanations for these observed disparities?
Because the etiology of BD is still unknown, the geographical distribution of BD and its fluctuation in the incidence of the disease in different regions is unexplained. For instance, the prevalence of BD in Japan has decreased from the duration of the Second World War to recently. The epidemiology part of the paper has been revised (highlighted in yellow).
3.The letter labels in the figure legends are too small, hindering readability. The quality of the figures and tables, in their current form, needs improvement.
The quality of the pictures, tables and their legends has been revised.
4.A substantial portion of the article summarizes the differential diagnosis between BD and ASO. Could the authors elaborate on how distinguishing between BD and ASO specifically based on pathological characteristics informs or influences clinical management decisions and treatment strategies?
About the management; according to my own experience, the clinical manifestation of BD patients who receive statins usually become worse that I had to stop giving them statins. I have addressed in one of my researches that BD patients might have mitochondrial dysfunction and this reason can explain this finding. I only prescribe statins in the older patients with atherosclerotic plaques. Also, ASO patients usually benefit from endovascular procedures and in BD patients it might be not successful or re-occlusion of the vessel after a few months could be expected. Beside all, for understanding the pathology and etiology of BD, we need to evaluate the similarities’ or differences of BD and its differential diagnosis. ASO is in the top list of BD differential diagnosis. Unless, we don’t know the exact pathophysiology of BD, our treatments would be blind. I have revised the conclusion part for mote clarification (highlighted in yellow).
Reviewer 2 Report
Comments and Suggestions for Authors
- The manuscript lacks definitions for the abbreviations VAS and ASO.
- In your article, you utilize the terms PAD and ASO, with a predominant emphasis on ASO. Could you clarify the distinctions between these terms within the context of your study?
- By referencing prior studies or reviews on Buerger's disease (BD), could you categorize the angiographic patterns observed in BD patients from early to late stages? It is plausible that angiographic findings may evolve depending on the disease stage in BD. The current manuscript lacks novelty; thus, establishing a classification of angiographic patterns across BD stages would significantly enhance its originality and impact.
Author Response
Dear Editor
It is our great pleasure to re-submit our revised manuscript entitled:” The Angiography Pattern of Buerger’s Disease: Challenges and Recommendations”.
I am very much thankful to the reviewers for their deep and thorough review and I do appreciate the constructive criticisms of the reviewers. I have revised our manuscript in the light of the useful suggestions and comments. I have addressed each of their concerns as outlined below. The revised parts are highlighted in yellow. I hope this time the revision has improved the manuscript to a level of reviewers’ satisfaction.
Reviewer 1:
1- The manuscript lacks definitions for the abbreviations VAS and ASO.
The abbreviation for ASO has been added and highlighted in yellow. However, about VAS, it is not an abbreviation. The full name of the foundation has been addressed and highlighted in yellow.
2- In your article, you utilize the terms PAD and ASO, with a predominant emphasis on ASO. Could you clarify the distinctions between these terms within the context of your study?
PAD means Peripheral arterial disease (including atherosclerotic, BD and other inflammatory disease) and ASO is atherosclerotic arterial disease. The PAD has been referred only once for describing Buerger’s disease as a kind of peripheral arterial disease. However, ASO has been considered as the main differential diagnosis of BD.
3- By referencing prior studies or reviews on Buerger's disease (BD), could you categorize the angiographic patterns observed in BD patients from early to late stages? It is plausible that angiographic findings may evolve depending on the disease stage in BD. The current manuscript lacks novelty; thus, establishing a classification of angiographic patterns across BD stages would significantly enhance its originality and impact.
Thanks for your great suggestion. It is a great idea that we should consider as an original research. However, according to the literatures, data for such categorizing is very limited. It has only been addressed that involvement of superficial femoral arteries or atherosclerotic plaques in the proximal arteries of the involved limb could be observed in the late stage of BD. Also, it seems development of corkscrew collaterals need time (highlighted in yellow). But for more details about pathophysiology of BD development, we should conduct an original research. The novelty of our review is describing “typical angiography pattern of BD” based on the frequency of the reported manifestations from 1950 until recently. We need to reach a consensus based on the literatures and experience of the authors for further data harmonization.
Round 2
Reviewer 1 Report
Comments and Suggestions for Authors
The author's revisions are good. I have no further comments.
Reviewer 2 Report
Comments and Suggestions for Authors
No comments.